# Effectiveness of Preoperative Chest Physiotherapy in Patients Undergoing Elective Cardiac Surgery, a Systematic Review and Meta-Analysis

**DOI:** 10.3390/medicina58070911

**Published:** 2022-07-08

**Authors:** Hadel Shahood, Annamaria Pakai, Rudolf Kiss, Bory Eva, Noemi Szilagyi, Adrienn Sandor, Zsofia Verzar

**Affiliations:** 1Doctoral School of Health Sciences, Faculty of Health Sciences, University of Pécs, 7624 Pécs, Hungary; 2Basic Health Sciences and Health Visiting, Institute of Nursing Sciences, Faculty of Health Sciences, University of Pécs, 7621 Pécs, Hungary; annamaria.pakai@etk.pte.hu; 3Heart Institute, Medical School, University of Pécs, 7624 Pécs, Hungary; kiss.rudi@gmail.com (R.K.); bjena78@gmail.com (B.E.); mimiszilagyi@gmail.com (N.S.); firerose05@gmail.com (A.S.); 4Institution of Nutritional Science and Dietetics, Faculty of Health Sciences, University of Pécs, 7621 Pécs, Hungary; verzar.zsofia@pte.hu

**Keywords:** preoperative chest physiotherapy, elective cardiac surgery, pulmonary functions, postoperative pulmonary complications

## Abstract

*Background and Objectives*: Patients undergoing cardiac surgery are particularly vulnerable for developing postoperative pulmonary complications (PPCs). This systematic review and meta-analysis aimed to evaluate the role of preoperative chest physiotherapy in such patients. *Materials and Methods*: All original articles that assessed patients undergoing elective cardiac surgery, with preoperative chest physiotherapy, and compared them to patients undergoing elective cardiac surgery, without preoperative chest physiotherapy, were included. Animal studies, studies conducted prior to the year 2000, commentaries, or general discussion papers whose authors did not present original data were excluded. Studies assessing physiotherapy regimens other than chest physiotherapy were also excluded. The search was performed using the following electronic resources: the Cochrane Central Register of Controlled Trials, the PubMed central database, and Embase. The included studies were assessed for potential bias using the Cochrane Collaboration’s tool for assessing the risk of bias. Each article was read carefully, and any relevant data were extracted. The extracted data were registered, tabulated, and analyzed using Review Manager software. *Results*: A total of 10 articles investigating 1458 patients were included in the study. The studies were published from 2006 to 2019. The populations were patients scheduled for elective CABG/cardiac surgery, and they were classified into two groups: the interventional (I) group, involving 651 patients, and the control (C) group, involving 807 patients. The meta-analysis demonstrated no significant differences between the interventional and control groups in surgery time and ICU duration, but a significant difference was found in the time of mechanical ventilation and the length of hospital stay, favoring the interventional group. A significant difference was shown in the forced expiratory volume in 1s (FEV1% _predicted_), forced vital capacity (FVC% _predicted_), and maximum inspiratory pressure (Pi-max), favoring the interventional group. *Conclusions*: This study is limited by the fact that one of the included ten studies was not an RCT. Moreover, due to lack of the assessment of certain variables in some studies, the highest number of studies included in a meta-analysis was the hospital stay length (eight studies), and the other variables were analyzed in a fewer number of studies. The data obtained can be considered as initial results until more inclusive RCTs are conducted involving a larger meta-analysis. However, in the present study, the intervention was proved to be protective against the occurrence of PPCs. The current work concluded that preoperative chest physiotherapy can yield better outcomes in patients undergoing elective cardiac surgery.

## 1. Background

In spite of the recent developments in the field of preoperative care, postoperative pulmonary complications (PPCs) remain a principal reason for operation-related morbidity and mortality [1]. PPCs are the respiratory system disorders that typically occur during the first postoperative week. These PPCs range from lung atelectasis to respiratory failure [2].

PPCs occur at a rate ranging from 1% to 23%. This wide variation is explained by the different risk factors related to the patients undergoing surgery [3]. The age of the patient is a risk factor, with higher risks being observed in healthy older patients [4]. Other factors include the lifestyle, habits, and cardio-respiratory health conditions of the patients [2].

In addition, the surgery and anesthesia impose factors that predispose the occurrence of PPCs [5]. Surgery may cause depressed lung function, as the surgical pain impacts normal breathing [6,7]. Anesthesia elicits adverse intraoperative, and, to a lesser extent, postoperative effects on pulmonary functions [8].

PPCs are predictors of the postoperative health outcomes of patients and increase the risk of admission to the intensive care units (ICUs), the prolongation of hospital stay length [9], and mortality [10].

Patients undergoing cardiac surgery are particularly vulnerable for developing PPCs. This is mainly attributed to the reduction in physical activity, the sternotomy incision, the cardiopulmonary bypass, and mechanical ventilation [11]. During cardiac surgery, the low ventilation–perfusion ratio elicits alveolar collapse, predominantly at the lung bases [7]. Additionally, the compression of the chest contributes to the development of atelectasis [6]. The postoperative residual effects of the neuromuscular blockade are also implicated in PPCs [8]. When prolonged intubation is needed for patients undergoing cardiac surgery, ventilator-associated pneumonia (VAP) can occur as a complication [11]. Moreover, the prolonged hospital stay increases the risk of nosocomial infections [9].

Chest physiotherapy is applied to minimize PPCs after CABG [12,13]. Despite the well-documented importance of postoperative physiotherapy [14], little is established on the value of preoperative intervention in patients undergoing cardiac surgery.

Preoperative interventions may be delivered to reduce PPCs in patients undergoing cardiac surgery [12]. These may involve the physiological optimization of the musculoskeletal system, such as inspiratory muscle training, breathing exercises [13], and exercise training [6], or improving the patient’s capacity to accommodate major surgery, such as relaxation therapy, and education [7].

Given the ongoing prevalence of postoperative morbidity and mortality, especially those attributed to the PPCs after elective cardiac surgery, it appears that standardization of the postoperative physiotherapy alone is not sufficient to preclude, or even minimize, the PPCs and their related morbidity and mortality. Thus, in this review, we tried to obtain any evidence derived, from research article review and analysis, suggesting potential benefits that could aid in the prophylaxis against the development PPCs.

The previously conducted pooled analyses either evaluated the effect of the preoperative intervention on patients undergoing any type of major surgery, or did not specify chest physiotherapy as the intervention procedure. Hence, we believe that, in order to fill this gap in the literature, a meta-analysis regarding the original articles addressing such issues is required.

Therefore, this systematic review and meta-analysis aimed to evaluate the role of preoperative chest physiotherapy in patients undergoing elective cardiac surgery. Analysis of the effect of preoperative chest physiotherapy on the incidence of PPCs and lung functions parameters, as a reflection of the overall respiratory system wellbeing and subsequently, the postoperative morbidity and mortality, was the goal of our research. Moreover, variations in the surgery duration, the length of stay in the ICU and hospital, and the time of mechanical ventilation were all objectives of our research, being aspects of the patient morbidity and the healthcare system spent cost.

## 2. Methods

### 2.1. Study Design

This is a systematic review and meta-analysis that was conducted in accordance with the Preferred Reporting Items for Systematic Reviews and Meta-Analyses (PRISMA) statement [15]. The review was registered in the Research Registry (reviewregistry1278).

### 2.2. Literature Search Strategy

The included studies were those evaluating the preoperative chest physiotherapy value in adult patients who undergo elective cardiac surgery. The search was performed using the electronic resources; the Cochrane Central Register of Controlled Trials, the PubMed central database, and Embase.

### 2.3. Selection Strategy and Criteria

The search was conducted with the restriction limiting results to original articles published from January 2000 to December 2021. The search was performed using the following keywords: “preoperative care” OR “preoperative” OR “preoperational” OR “pre-habilitation” OR “pre-habilitation” OR “before operation” OR “ before surgery” AND “coronary artery disease” OR “CAD” AND “chest” OR “respiratory” OR “lung” OR “pulmonary” AND “physiotherapy” OR “physical therapy” OR “muscular training” OR “muscle training” OR “muscle exercise” OR “muscular exercise” OR “muscle strength” AND “cardiac surgery” OR “open cardiac surgery” OR “open heart surgery” OR “heart surgery” OR “coronary artery bypass graft” OR “coronary artery bypass grafting” OR “CABG” AND “postoperative” OR “following operation” OR “after” OR “after cardiac operation” AND “pulmonary complications” OR “lung complications” OR “lung impairment” OR “respiratory failure” OR “respiratory impairment” OR “impaired respiratory functions” OR “impaired lung functions.” The search was performed by two independent reviewers (the first and second authors). Then articles were matched and screened to ensure eligibility. The search strategy in each data base was described in Appendix A.

#### 2.3.1. Inclusion Criteria

Original articles available in English were included. According to PICO, we included the studies meeting the following criteria: study design—all original articles that included randomized controlled trials or observational studies from 2000 until conducting the analysis (mid-2021), participants—patients undergoing elective cardiac surgery, intervention—preoperative chest physiotherapy, control—patients undergoing elective cardiac surgery without preoperative chest physiotherapy, outcome measures—the effect of intervention on PPCs and any other effect.

#### 2.3.2. Exclusion Criteria

Animal studies, studies completed prior to the year 2000, commentaries, or general discussion papers whose authors did not present original data were excluded. Studies assessing physiotherapy regimens other than chest physiotherapy and those applying postoperative physiotherapy programs other than the routine therapies were also excluded.

### 2.4. Data Extraction, Data Collection, and Analysis

Each article was read carefully and any relevant data were extracted (including the study setting, design, research questions, sample size, patients’ demographic data, medical history, baseline preoperative data, type and details regarding the intervention, description of the intervention; type, time, duration, rate, and the used device, lung function tests, muscle strength, operative events, length of hospital stay, the occurrence of postoperative pulmonary complications, and the study conclusions). The extracted data were registered, tabulated, and analyzed.

### 2.5. Bias

Methodological quality check lists were used as tools for bias risk assessment. The included studies were assessed for potential bias using the Cochrane Collaboration’s tool for assessing the risk of bias.

### 2.6. Summary Measures

The primary outcomes were the incidence of postoperative complications and the changes in the lung function parameters, and the secondary outcomes were the surgery duration, the length of stay in the ICU and hospital, and the time of mechanical ventilation.

The assessed lung function parameters were:FEV1% _predicted_: forced expiratory volume (FEV1%) of the patient divided by the average FEV1%.FVC% _predicted:_ forced vital capacity (FVC%) of the patient divided by average FVC%.Pi-max: maximum inspiratory pressure.

Data about the ongoing RCTs related to the study topic was evaluated and described in the Discussion section.

### 2.7. Statistical Analysis

The retrieved data were presented as mean and standard deviation (SD) for numerical data, and frequency and percentage for categorical data.

The meta-analysis and bias assessment were accomplished using the Review Manager software (RevMan version 5.4, the Cochrane Collaboration, London, UK). Dichotomous data were expressed as a risk ratio, with 95% confidence intervals (CIs) to compare intervention and control groups at the study level. For continuous outcomes, the mean differences in effects between the intervention and control groups were computed at the study level and pooled into weighted mean differences (WMDs).

## 3. Results

The search of the electronic resources first yielded a total of 24,106 records. After duplication adjustment, the search provided 1123 results. Based on the title, 898 publications were excluded. Then, after checking abstracts, another 199 publications were found not to meet the eligibility criteria, so they were further excluded. After checking the full texts, an additional 16 articles were excluded. Thus, 10 studies were finally eligible for this systematic review (Figure 1).

The included articles were published from 2006 to 2019. The populations were patients scheduled for elective CABG/cardiac surgery in the Netherlands, Turkey, Taiwan, Iran, Brazil, Pakistan, and China (Table 1).

The included studies had a total population number of 1458. They were classified into two groups: the interventional (I) group, involving 651 patients, and the control (C) group involving, 807 patients. Five studies investigated a sample size of <100 [16,17,18,19,20], and the others investigated a sample size of >100 [21,22,23,24,25].

All patients were adults, with a mean age of 60.36 ± 12.78. Male predominance was noted, as 1019 (69.9%) patients were males, while only 439 patients were females. No significant difference was found among all the included studies regarding either age or male/female ratio (*p* > 0.05) (Table 2).

The percentages of smokers ranged from 25% to 70% in the review studies, BMI ranged from 25.66 to more than 30, and the comorbidities were mainly diabetes mellitus, hypertension, and hyperlipidemia. Both groups in all studies were matched according to the prevalence of risk factors and the comorbidities (Table 3).

Concerning the type of preoperative intervention in the included studies, some used respiratory training protocols, with an incentive spirometer [16,20,23], one study combined incentive spirometer with a threshold loading device [24], and others used threshold loading devices for chest physiotherapy [16,17,19,20,22,25] (Table 4).

The time frame for preoperative intervention application differed considerably among the included studies, ranging from 5 days to 10weeks. The frequency of performing the interventional program ranged from twice a day [25], to three times every two weeks [18]. The duration of training sessions ranged from 20 [16,21,22,25] to 60 min [18] (Table 4).

Preoperatively, the control groups underwent the usual management [16,18,19,21,22] or usual management in addition to 1 day of chest physiotherapy [24], limbs and trunk mobilization [17], or abdominal breathing training [25] (Table 5).

Postoperatively, both groups received chest physiotherapy and mobilization schemes [16,17,18,20,21,23,24,25] or physiotherapy, as required [19] (Table 5).

Regarding study outcomes, the primary outcome was the existence of PPCs [18,21,22,25], the occurrence of adverse events, in addition to the degree of patient satisfaction and motivation [16], the inspiratory muscle strength [17,19] spirometry parameters [20], postoperative oxygenation [23], and the quality of life [24]. The postoperative stay length was the secondary outcome in 5 studies [16,19,21,22,25] (Table 6).

In all studies, the basic preoperative pulmonary functions, respiratory muscle test parameters, and ABGs were comparable in the two groups.

The pooled analysis revealed no significant differences between the interventional and control groups in the surgery time (Figure 2) and the ICU duration (Figure 3) (*p* = 0.84 and 0.92, respectively), with no heterogeneity in the results (*p* = 0.06 and 0.62, respectively).

There were significant differences between the intervention and control groups in the duration of mechanical ventilation (Figure 4) and the length of hospital stay (Figure 5) (*p* < 0.001), favoring the interventional group. The pooled mean differences between groups were 0.76 h and 1.02 days, respectively. The absence of heterogeneity in the meta-analyses of the length of hospital stay (*p* = 0.1) grants credibility to the results.

The meta-analyses revealed significant differences between the interventional and control groups in the FEV1%_predicted_ (Figure 6), FVC% _predicted_ (Figure 7), and Pi-max (Figure 8) (*p* < 0.05), favoring the interventional group. The pooled mean differences were 3.7%, 10.17%, and 17.25 cm H_2_O, respectively.

The PPCs meta-analysis demonstrated that the intervention had a protective effect on the occurrence of PPCs. The pooled risk ratio was shown to be 47%, with a 95%CI of 36–62%. The overlap between a part of the CI that was shown in the pooled estimates reflected the absence of statistical heterogeneity (I_2_ = 0%, *p* = 0.73) (Figure 9).

A funnel plot was constructed (Figure 10), revealing the symmetry in results, as all the involved studies lay within the confidence interval, with a rather symmetric pattern, ensuring the absence of heterogeneity in the results.

Regarding the QoL results in this systematic review, 3of the 10 included studies introduced at least one QoL parameter, with different scales used by the authors. Vakenet et al. (2017) demonstrated that Qol differences were less in the interventional group compared to the control group [24], yet with non-statistical significance (*p* > 0.05); the work of Savci et al. (2011) also failed to reveal a statistically significant difference between both groups in the assessed physical component of QoL (*p* > 0.05) [17]. In contrast, Tung et al. (2012) noticed an intervention-related significant improvement in the general QoL (*p* < 0.001) [18]. Anxiety and depression were evaluated in one study [17], which showed that their expressions were lower in the intervention group. However, this difference was significant only in the anxiety component (*p* < 0.05).

A summary of the study outcomes is demonstrated in Table 7.

When examining the conclusions reached by the included studies, there was unanimous agreement on the importance and significance of preoperative chest physiotherapy in patients undergoing elective cardiac surgery.

The critical assessment graph and a summary of the risks of bias within each study are shown in Figure 11 and Figure 12.

## 4. Discussion

The studies included in this review explored variable methods for preoperative chest physiotherapy. Several strategies are created to intervene in an attempt to prevent PPC development. Interventions may be preoperative, to adjust the physiology of respiration, or intraoperative and postoperative, to minimize the adverse events of surgery and anesthesia [26]. Despite this, there are no established guidelines for preoperative protocols of management. Even if they are present, they are outdated or infrequent [27]. This lack of consensus leads to considerable variation in clinical practice [28].

The current study pooled analysis demonstrated that there was no effect of the intervention on the surgery time or the ICU stay duration, while it favorably affected the mechanical ventilation and the length of hospital stay.

The association of preoperative chest physiotherapy with shorter hospital stay was also documented in previous studies [29,30]. Nardi et al. (2019) observed that the length of the postoperative hospital stay in the group that had preoperative training was reduced compared to the control group, but without a statistically significant difference [13].

The short hospital stay affords the patients the chance to continue the recovery in their familiar home environment, saving the hospital resources for new patients to receive health care services [31,32].

In contrast with the findings of our study, the previous meta-analyses did not reveal a significant difference between the interventional and control groups in the mechanical ventilation time [33,34,35]. However, we can state that the significant difference found in our study could be a quasi-significance, due to the heterogeneity found in the results.

The meta-analysis of this study showed that there was a significant difference between the interventional and control groups in the pulmonary functions, including FEV1% _predicted_, FVC% _predicted_, and Pi-max, favoring the interventional group. However, only the FEV1%_predicted_ showed homogenous results.

The Pi-max was the most commonly tested lung function parameter in the included studies. Pi-max reflects the inspiratory muscles’ functional capacity and has been adopted as a reliable indicator for the weaning from mechanical ventilation in many hospitals [36]. In our meta-analysis, the evidence of the intervention’s improving effect was weak. Within the same context, in the meta-analysis conducted by Marmelo et al. (2018), the authors found significant improvement of the Pi-max related to the intervention [33]. On the other hand, Katsura et al. (2015) reported no statistically significant effect in a three-articles meta-analysis, in spite of the fact that a tendency toward a favoring effect was found in all three articles [35].

The current meta-analysis revealed that intervention proved to be protective against the existence of PPCs.

Consistent with our findings, the recent meta-analysis conducted by Odor et al. (2020) disclosed evidence of the prophylactic effect of preoperative physiotherapy against the occurrence of PPCs [28]. Our findings were also congruent with the most recent meta-analysis conducted by Rodrigues et al. (2021), which demonstrated that preoperative chest physiotherapy (breathing interventions) helped to improve postoperative respiratory performance in patients undergoing cardiac surgery. Moreover, the authors concluded that such interventions reduced PPCs and the length of hospital stay [37]. Other previous studies affirmed the effect of preoperative intervention on PPCs. This was investigated in patients who underwent oncologic thoracic surgeries [38], cardiac [33,38] intra-abdominal [39], and cardiac and abdominal surgery [30,40]. In these meta-analyses, a total of 31 studies reported decreased PPCs in the interventional group, while only 8 did not find this relationship. The study of Kamarajah et al. (2019) highlighted that pre-habilitation improved rates of morbidity, including for PPCs, and overall complications after both major abdominal and cardiothoracic surgery [41].

The recently published work of our group showed supporting findings [42]. Our RCT demonstrated overall significant postoperative improvements in lung function and oxygen saturation in the intervention group compared with the control group. An earlier RCT conducted by Sweity et al. (2021) to assess the effect of preoperative incentive spirometry was compatible with our findings, as the study showed a significant difference between the interventional and control groups in the incidence of postoperative atelectasis, mechanical ventilation duration, and hospital LOS. The median of the amount of arterial blood oxygen and oxygen saturation was significantly improved in the intervention group [43].

The QoL outcome exhibited heterogeneity in the measured scale and the obtained results. The variability in the used methods makes it difficult to obtain a consensus about the results [44,45]. The QoL variables’ interpretation had to be considered in view of the individual results of the articles assessing this outcome, since we could not conduct a meta-analysis in view of the heterogeneity in the quantification scales. In this regard, Valkenet et al. (2017) reported that the intervention group showed less reduction in QoL values than the control group, but without a statistical significance [24]. Savci et al. (2011) also failed to reveal a statistically significant difference between either groups in the assessed physical component of QoL [17], while Tung et al. (2012) proved a significant improvement in the general QoL [18].

The lack of evidence found in this review is in accordance with Marmelo et al. (2018) [33], Katsura et al. (2015) [35], and Santa-Mina et al. (2014) [31], who reported similar findings. Hulzebos et al. (2012) actually found better QoL results in the control groups [34].

Anxiety and depression was evaluated in the study of Savci et al. (2011) [17], which showed that anxiety and depression tendencies were lower in the intervention group than in the control group. However, this difference was significant only in the anxiety component. In congruence, an earlier study demonstrated that the patients who were preoperatively educated, with guidance on the physiotherapeutic ventilation training, exhibited reduced anxiety levels compared with those did not receive this guidance [46].

It is worth noting that all the included patients were those scheduled for CABG, except for those in the studies of Tung et al. (2012) and Chen et al. (2019), which also included patients with valve surgeries. Both types of surgeries are open heart surgeries, with the indicated general anesthesia, median sternotomy incision, cardiopulmonary bypass, and mechanical ventilation. All of these are factors implicated in the predisposition to PPCs and hence, both types of surgeries were included in the analysis.

Working to prevent or reduce the incidence of pulmonary complications occurring in patients after heart surgery is a major goal among health workers. To achieve this goal, it is recommended to educate patients about how important is to learn the physical therapy techniques. These techniques could help improve the respiratory functions and promote the expansion of the lungs, identifying patients at high-risk for the development of pulmonary complications after surgery. In this vein, physical therapy is highly regarded among the basic treatments and should be offered to the patients in intensive care units. This study confirms the potential performance of a rehabilitation program before cardiac surgery, recommending its availability to all patients, if possible, in order to make the post-operative period less traumatic, and to facilitate a faster functional recovery.

## 5. Strengths and Limitations

This study is strengthened by including a meta-analysis in addition to the systematic review of the included studies. Moreover, the pooled analysis included a large number of patients, thus yielding a rather firm conclusion. This study is limited by the fact that one of the included ten studies was not an RCT. Moreover, due to the lack of assessment of certain variables in some studies, the highest number of studies included in a meta-analysis was in the hospital stay length (eight studies), and the other variables were analyzed in a fewer number of studies. The data obtained can be considered as initial results until more inclusive RCTs are conducted involving a larger meta-analysis.

## 6. Conclusions

The current work concluded that preoperative chest physiotherapy can yield better outcomes in patients undergoing elective cardiac surgery.

The meta-analysis demonstrated no significant differences between the interventional and control groups in the surgery time and the ICU duration, but a significant difference in the time of mechanical ventilation and the hospital stay length, favoring the interventional group. A significant difference was shown in the FEV1% _predicted_, FVC% _predicted_, and Pi-max, favoring of the interventional group. The most notable significance was shown in the analysis of hospital stay length and the FEV1% _predicted_. The intervention was proved to be protective against the occurrence of PPCs.

## Figures and Tables

**Figure 1 medicina-58-00911-f001:**
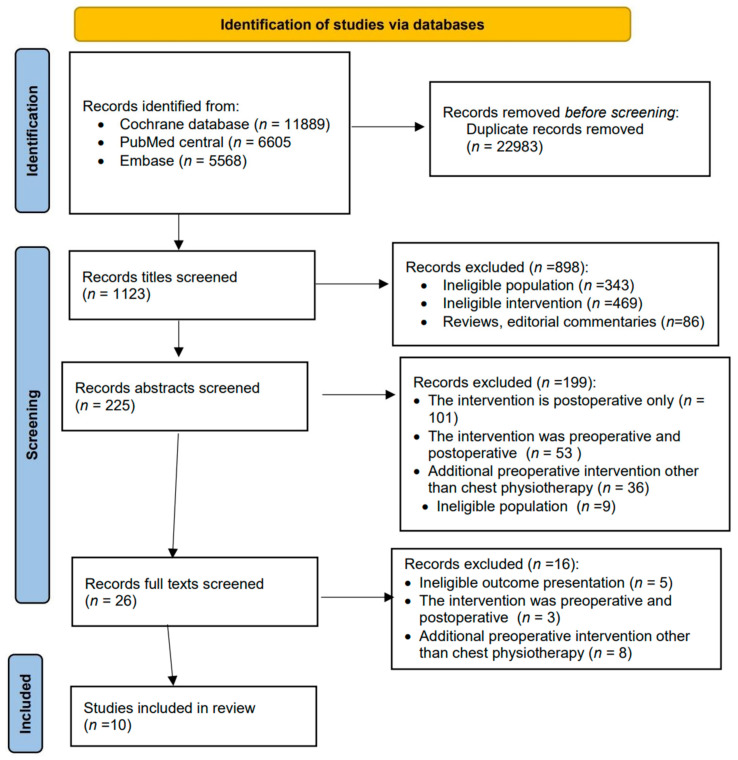
PRISMA study selection flow chart.

**Figure 2 medicina-58-00911-f002:**
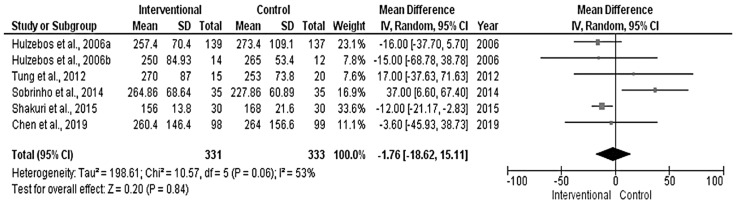
Forest plot evaluating surgery time in the included studies.

**Figure 3 medicina-58-00911-f003:**
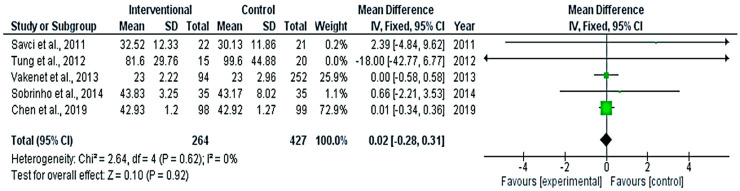
Forest plot evaluating ICU duration stay in the included studies.

**Figure 4 medicina-58-00911-f004:**
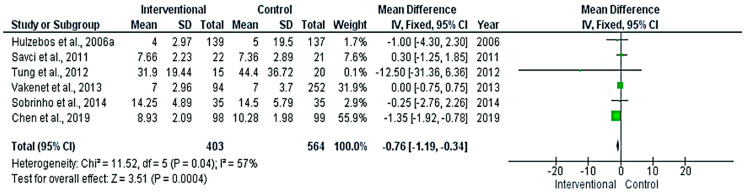
Forest plot evaluating mechanical ventilation duration in the included studies.

**Figure 5 medicina-58-00911-f005:**
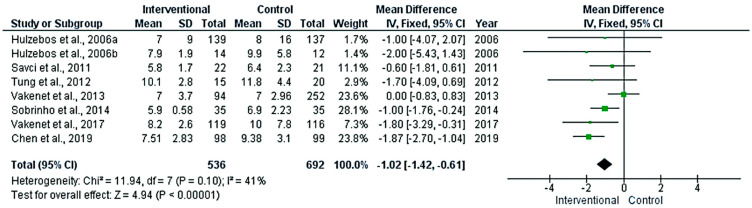
Forest plot evaluating hospital stay length in the included studies.

**Figure 6 medicina-58-00911-f006:**
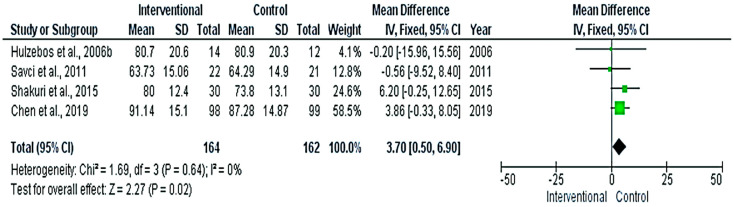
Forest plot evaluating FEV1% _predicted_ in the included studies.

**Figure 7 medicina-58-00911-f007:**
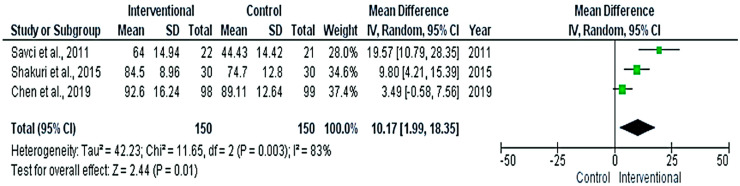
Forest plot evaluating FVC% _predicted_ in the included studies.

**Figure 8 medicina-58-00911-f008:**
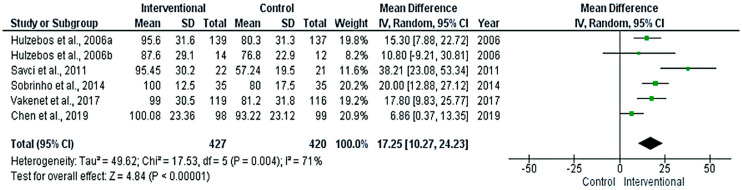
Forest plot evaluating Pi-max in the included studies.

**Figure 9 medicina-58-00911-f009:**
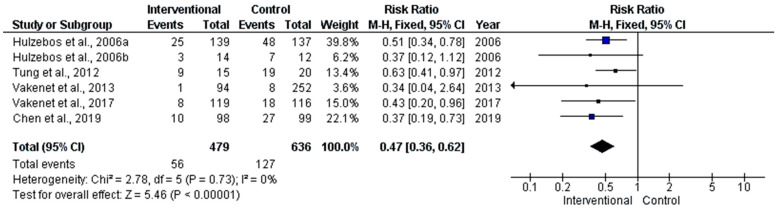
Forest plot evaluating PPCs in the included studies.

**Figure 10 medicina-58-00911-f010:**
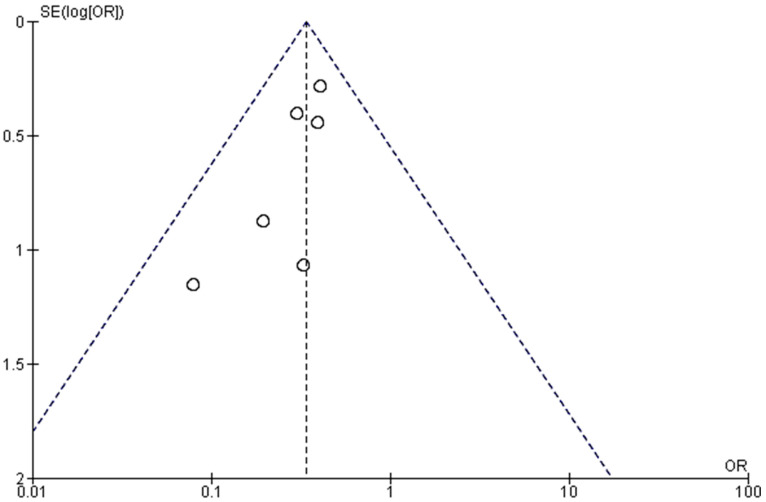
Funnel plot for the PPCs incidences reported by the studies.

**Figure 11 medicina-58-00911-f011:**
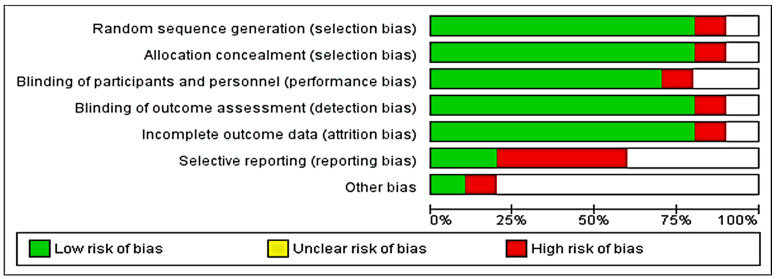
Risk of bias graph for the included studies.

**Figure 12 medicina-58-00911-f012:**
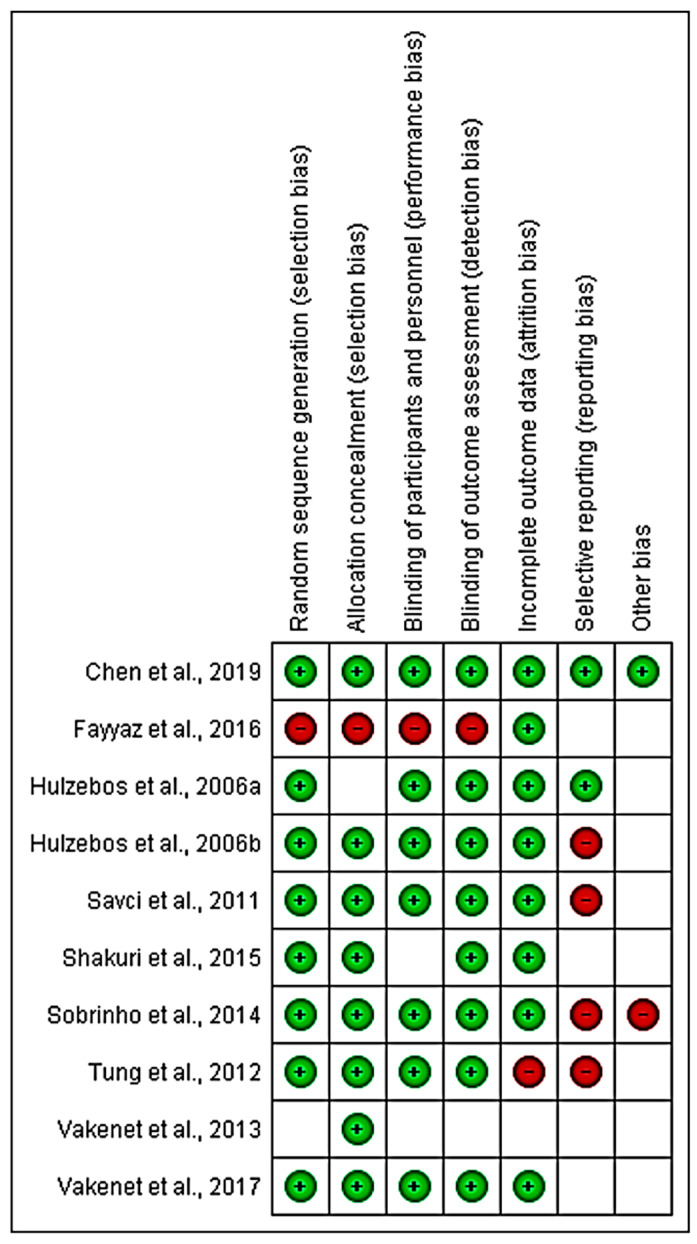
Risk of bias summary for the included studies. A square with a green circle means low risk, a square with a red circle means high risk, and an empty square indicates unclear risk.

**Table 1 medicina-58-00911-t001:** The location and design of the included studies.

Study	Study Place	Study Design	Objected Patients
Hulzebos et al., 2006a	Netherlands	RCT	Patients planned for elective CABG
Hulzebos et al., 2006b	Netherlands	RC pilot study	Patients planned for elective CABG, with high risk to develop PPCs.
Savci et al., 2011	Turkey	RCT	Patients planned for elective CABG.
Tung et al., 2012	Taiwan	RC pilot study	Patients planned for CABG and/or valve surgeries.
Vakenet et al., 2013	Netherlands	Observational cohort study	Patients planned for elective CABG, with high risk to develop PPCs.
Sobrinho et al., 2014	Brazil	RCT	Patients planned for elective CABG.
Shakuri et al., 2015	Iran	RCT	Patients planned for open cardiac surgery.
Fayyaz et al., 2016	Pakistan	RCT	Patients planned for elective CABG.
Vakenet et al., 2017	Netherlands	2ry analyses of RCT	Patients planned for elective CABG.
Chen et al., 2019	China	RC pilot study	Patients planned for CABG and/or valve surgeries.

RCT: randomized controlled trial; CABG: coronary artery bypass graft; PPCs: postoperative pulmonary complications.

**Table 2 medicina-58-00911-t002:** A summary of the sociodemographic data included in the studies.

Study	N (Total: I–C)	Mean Age (I–C)	Male % (I–C)
Hulzebos et al., 2006a	276: 139–137	66.5–67.3	77.7–78.1
Hulzebos et al., 2006b	26: 14–12	70.14–70.5	50–50
Savci et al., 2011	43: 22–21	62.82–57.48	86.4–90.5
Tung et al., 2012	35: 15–20	52.5–54.7	93.3–70
Vakenet et al., 2013	346: 94–252	66.8–68.4	61.7–68.3
Sobrinho et al., 2014	70: 35–35	58.9–61.4	65.7–82.9
Shakuri et al., 2015	60: 30–30	54.4–59.3	63.3–66.1
Fayyaz et al., 2016	170: 85–85	39.44–39.33	--
Vakenet et al., 2017	235: 119–116	66–67.5	78.2–80.2
Chen et al., 2019	197: 98–99	61.86–61.86	74.5–68.7
Total	1458: 651–807	60.36	69.9

I: Intervention group; C: Control group.

**Table 3 medicina-58-00911-t003:** The risk factors and comorbidities of patients included in the studies.

Study	Smoking % (I–C)	BMI Mean (I–C)	Comorbidity % (I–C)
Hulzebos et al., 2006a	32.4–38	28.3–28.1	HTN: 57–54.5 DM: 43.9–32.8 COPD: 19.4–21.9 Hyperlipidemia: 25.9–26.3
Hulzebos et al., 2006b	29–25	26.13–28.32	DM: 14–25 COPD: 43-17
Savci et al., 2011	70.95–71.62	27.49–25.73	HTN, DM, hyperlipidemia, alcohol consumption, inactivity, and family history
Tung et al., 2012	60–70	27.8–26.3	<3: 86.7–80 >3: 13.3–20
Vakenet et al., 2013	--	≥30 (%): 27.7–27.8	DM: 34–57.1
Sobrinho et al., 2014	67–67	27.08–26	--
Shakuri et al., 2015	30–33.3	26.8–27.7	DM: 36.7–26.6
Fayyaz et al., 2016	--	28.36–26.20	--
Vakenet et al., 2017	34.5–36.2	28.6–28.1	HTN: 58.8–44 DM: 42.9–30.2
Chen et al., 2019	44.9–37.4	26.07–25.66	DM: 25.5–27.3 HTN: 56.1–67.7 Hyperlipidemia: 5.1–3.0

I: intervention group; C: control group; HTN: hypertension; DM: diabetes mellitus; COPD: chronic obstructive pulmonary disease.

**Table 4 medicina-58-00911-t004:** The type and description of interventions in the included studies.

Study	Preoperative Intervention	Time Frame/Frequency	Session Duration	Used Tool
Hulzebos et al., 2006a	IMT; incentive spirometry, once a week with supervision by a physical therapist.	2–10 weeks/daily	20 min	Threshold IMT^®^
Hulzebos et al., 2006b	IMT; incentive spirometry, once a week with supervision by a physical therapist.	2–4 weeks/daily	20 min	Threshold IMT^®^
Savci et al., 2011	IMT under the supervision of a physical therapist.	5 days/daily	30 min	Threshold IMT (Respironics, Pittsburg, PA, USA).
Tung et al., 2012	Individualized, tailored exercises—PIEP. The PIEP was set at a low intensity, i.e., achieving 50–60% maximal oxygen consumption (VO_2_max) for this population, by an expert panel.	2 weeks/once or twice a weak (3 times)	40–60 min	Cycle ergometer, spirometer, SF-36
Vakenet et al., 2013	Unsupervised IMT program at home.	2 weeks/daily	20 min/day	Threshold IMT, (Respironics, New Jersey, PA, USA).
Sobrinho et al., 2014	Breathing exercises.	Daily till surgery, once a day	Not specified	Threshold—IMT^®^
Shakuri et al., 2015	Exercises and auxiliary activities for extension and rotation of thoracic vertebrae, breathing exercises, exercises to expand lung lobes, aerobic exercises at a constant low speed for all the patients.		25 min	flow-incentive spirometer-based (Respiflow™ FS)
Fayyaz et al., 2016	Incentive spirometry.	--	--	--
Vakenet et al., 2017	IMT; incentive spirometry; education.	2 weeks/daily		
Chen et al., 2019	IMT.	5 days/twice a day	20 min	Threshold IMT device (HS730-010; Philips Respironics, Pittsburgh, PA, USA).

IMT: inspiratory muscle training; PIEP: preoperative individualized exercise prescription.

**Table 5 medicina-58-00911-t005:** Description of the preoperative protocols.

Study	Preoperative Control Group Management	Postoperative Both Groups Management
Hulzebos et al., 2006a	Care as usual the day before surgery (i.e., instruction on deep breathing maneuvers, coughing, and early mobilization).	Incentive spirometry, chest physical therapy, and mobilization scheme after operation.
Hulzebos et al., 2006b	Care as usual the day before surgery (i.e., instruction on deep breathing maneuvers, coughing, and early mobilization).	Incentive spirometry, chest physical therapy, and mobilization scheme after operation.
Savci et al., 2011	Mobilization, active exercises of upper and lower limbs, chest physiotherapy.	Chest physical therapy and mobilization scheme after operation.
Tung et al., 2012	Care as usual the day before surgery (i.e., instruction on deep breathing maneuvers, coughing, and early mobilization).	Incentive spirometry, chest physical therapy, and mobilization scheme after operation.
Vakenet et al., 2013	Received usual care (no IMT).	Incentive spirometry, chest physical therapy, and mobilization scheme after operation.
Sobrinho et al., 2014	Only routine ward guidelines before surgery.	Physical therapy as needed by staff physiotherapy service.
Shakuri et al., 2015	--	Incentive spirometry, chest physical therapy, and mobilization scheme after operation.
Fayyaz et al., 2016	--	--
Vakenet et al., 2017	Care as usual the day before surgery (i.e., instruction on deep breathing maneuvers, coughing, and early mobilization).	Incentive spirometry, chest physical therapy, and mobilization scheme after operation.
Chen et al., 2019	Both groups received both usual care (i.e., education, coughing and early mobilization) and abdominal breathing training before the surgery.	Chest physical therapy and mobilization scheme after operation.

IMT: inspiratory muscle training.

**Table 6 medicina-58-00911-t006:** The outcomes of the included studies.

Study	Outcome Measure
Hulzebos et al., 2006a	The primary outcome: the incidence of PPCs. The secondary outcome was duration of postoperative hospitalization.
Hulzebos et al., 2006 b	Primary outcome variables: the occurrence of adverse events and patient satisfaction and motivation. Secondary outcome variables: postoperative pulmonary complications and length of hospital stay.
Savci et al., 2011	Inspiratory muscle strength (cmH_2_O). Quality of life was assessed using the Nottingham Health Profile. Anxiety and depression were measured using the Hospital Anxiety and Depression Scale (HADS).
Tung et al., 2012	Pulmonary complication-related parameters. Quality of life assessment using Short Form 36-Health Survey (SF-36).
Vakenet et al., 2013	The primary outcome measure: postoperative pneumonia. The secondary outcome measures: ventilation time, postoperative length of stay (LOS) in the intensive care unit (ICU), and total LOS.
Sobrinho et al., 2014	The respiratory muscle strength, pulmonary volumes, and duration of hospital stay after surgery.
Shakuri et al., 2015	Spirometry parameters; ABG parameters.
Fayyaz et al., 2016	Postoperative oxygenation.
Vakenet et al., 2017	Quality of life assessment using Short Form 36-Health Survey (SF-36).
Chen et al., 2019	The primary outcome variable: the occurrence of postoperative pulmonary complications. The secondary outcome variables: inspiratory muscle strength, lung function, and length of hospitalization.

PPCs: postoperative pulmonary complications; ABG: arterial blood gases.

**Table 7 medicina-58-00911-t007:** The outcomes summary of the included studies.

Study	Outcome Measure
Hulzebos et al., 2006a	The primary outcome: a statistically significant difference was found between the two groups in the incidence of PPCs (*p* = 0.02). The secondary outcome: a statistically significant difference was found between the two groups in the LOS (*p* = 0.02).
Hulzebos et al., 2006b	Primary outcome: the feasibility of the intervention was good. No adverse events were reported. A statistically significant difference was found between the two groups in the satisfaction scores and the muscle strength. Secondary outcome: a statistically significant difference was found between the two groups in the incidence of PPCs, but not in the LOS (*p* = 0.24)
Savci et al., 2011	Statistically significant difference was found between the two groups in the MIT, the improvement in QoL, and the anxiety score of HADS.
Tung et al., 2012	Significant reduction in the non-invasive ventilator (*p* = 0.012), the time to ambulance, and the PPCs, and better general health scores were shown in the intervention group
Vakenet et al., 2013	Statistically significant difference was found between the two groups in the incidence of PPCs, but not in the LOS
Sobrinho et al., 2014	Statistically significant difference was found between the two groups in the MIT and LOS.
Shakuri et al., 2015	Significant changes in predicted FVC, PEF, and PCO2 concentration in the interventional group compared to the control group.
Fayyaz et al., 2016	Significant postoperative improvement of PO2 and PCO2 in the interventional group compared to the control group.
Vakenet et al., 2017	No significant differences in change of QoL scores over time were found between the intervention and control groups.
Chen et al., 2019	Statistically significant difference was found between the two groups in the incidence of PPCs, the MIT, and LOS.

PPCs: postoperative pulmonary complications; LOS: length of stay; QoL: quality of life; HADS: Hospital Anxiety and Depression Scale.

## Data Availability

Data will be available upon request.

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
