# Peer review of "Effectiveness of Preoperative Chest Physiotherapy in Patients Undergoing Elective Cardiac Surgery, a Systematic Review and Meta-Analysis"

_medicina, 2022, doi:10.3390/medicina58070911_

Round 1
Reviewer 1 Report
The authors of the manuscript entitled “Effectiveness of Preoperative Chest Physiotherapy in Patients undergoing Elective Cardiac Surgery,A Systematic Review and Meta-Analysis” aims to evaluate the role of preoperative chest physiotherapy in such patients.
In my opinion, this paper should be revised before publication. My concerns about the paper are as follows. Some suggestions or questions to improve the quality of the work are presented:
Abstract
Abbreviations should be described in the abstract the first time that appear.
Abstract should be improved following the PRISMA Guideliness.
Methods
Specific search strategy used for each database should be provided as supplementary or additional material.
Methodological quality of the included studies could be evaluated added to the risk of bias.
It would be interesting providing information about the ongoing RCT related to the topic.
Results
The original PRISMA flow chart should be used. It should be indicated the reasons of exclusions.
Quality of figures should be improved, text appear distorted
Tables need a footnote where abbreviations should be described.
Please revise spaces in the text.
Author Response
Medicina 2021, 57, x
Response to the reviewers’ comments
First of all, we would like to deeply thank the chief editor, the editorial office and the reviewers for the appreciated time and valuable recommendations. We followed all the suggested revision points. Reviewer 1 comments-based modification in the article was marked in blue, while Reviewer 2 comments-based modification was underlined.
|
Reviewer Number |
Original comments of the reviewer |
Reply by the author(s) |
|
1 |
1. Abbreviations should be described in the abstract the first time that appear.
|
Many thanks for your valuable comments and esteemed support. The abbreviations were described as required.
|
|
1 |
2. Abstract should be improved following the PRISMA Guidelines.
|
The abstract was revised and edited to follow the PRISMA guidelines. |
|
1 |
3. Specific search strategy used for each database should be provided as supplementary or additional material.
|
A supplementary file was added as recommended. |
|
1 |
4.Methodological quality of the included studies could be evaluated added to the risk of bias. |
Actually, this was performed and we added this clarification in the section of bias assessment. |
|
1 |
5. It would be interesting providing information about the ongoing RCT related to the topic. |
Data about the ongoing RCTs related to the study topic was evaluated and described in the discussion section as advised. |
|
1 |
6. The original PRISMA flow chart should be used. It should be indicated the reasons of exclusions.
|
The recommended chart was used and the reasons for exclusion were added. |
|
|
7. Quality of figures should be improved, text appear distorted |
The quality of figures was improved as suggested. |
|
|
8. Tables need a footnote where abbreviations should be described. |
Tables footnotes were added. |
|
1 |
9. Please revise spaces in the text. |
Spaces were revised and corrected as required. |
Reviewer 2 Report
The authors present an interesting study, to which I have several comments and recommendations.
Introduction - in the last part of the Backgroud chapter I recommend to better, more clearly express the goal of the whole work, I consider the above formulations to be too general
Methods - ll. 79-81 does not need to be stated, information on registration is sufficient
Inclusion criteria - the authors intend to include "all original articles", (l. 104) here it is necessary that the level of studies is clearly defined, I lack information about years, only in exclusion criterias there is information about 2020
Summary measures – l. 126 authors speak of a lung function parameters, these parameters would need to be defined in advance
Tables - it is necessary to add a legend, explanation of abbreviations for all tables
Figure 12: it is necessary to explain the symbols used
Citing sources in the text - it would be better to cite sources in parentheses instead of the superscript used
The presented manuscript works with a very small number of studies, the authors are aware of this fact in limitations. Perhaps it would be beneficial for the whole manuscript in the discussion chapter to elaborate more in depth on the basis of the results obtained and the impact on practice.
Despite the fact that the number of studies in the search is small, I see the presented work as beneficial and, above all, as a suitable basis for research in this area.
Author Response
Medicina 2021, 57, x
Response to the reviewers’ comments
First of all, we would like to deeply thank the chief editor, the editorial office and the reviewers for the appreciated time and valuable recommendations. We followed all the suggested revision points. Reviewer 1 comments-based modification in the article was marked in blue, while Reviewer 2 comments-based modification was underlined.
|
2 |
1. in the last part of the Backgroud chapter I recommend to better, more clearly express the goal of the whole work, I consider the above formulations to be too general
|
Thanks a lot for your appreciated time and valuable recommendations. The goal of work was expressed more clearly as recommended. |
|
2
|
2. Methods - ll. 79-81 does not need to be stated, information on registration is sufficient
|
The described link was removed as recommended. |
|
2 |
3. Inclusion criteria - the authors intend to include "all original articles", (l. 104) here it is necessary that the level of studies is clearly defined, I lack information about years, only in exclusion criterias there is information about 2020. |
This was more clearly described. |
|
2 |
4. ‘Summary measures – l. 126 authors speak of a lung function parameters, these parameters would need to be defined in advance. |
The parameters were defined as suggested.
|
|
2 |
5. Tables - it is necessary to add a legend, explanation of abbreviations for all tables. |
Tables’ ligands with abbreviations explanation were added as required. |
|
2 |
6. Figure 12: it is necessary to explain the symbols used |
The symbols were explained as required. |
|
2 |
7. Citing sources in the text - it would be better to cite sources in parentheses instead of the superscript used
|
The source were cited in the text in parentheses instead of superscript as recommended. |
|
2
|
8. The presented manuscript works with a very small number of studies, the authors are aware of this fact in limitations. Perhaps it would be beneficial for the whole manuscript in the discussion chapter to elaborate more in depth on the basis of the results obtained and the impact on practice.
|
The impact on practice and clinical implications of our research were described at the end of the discussion as suggested.
|
|
2 |
9. Despite the fact that the number of studies in the search is small, I see the presented work as beneficial and, above all, as a suitable basis for research in this area. |
Many thanks sir for your valuable comments and esteemed support.
|
Round 2
Reviewer 2 Report
The authors have made changes that have contributed to the improvement of the manuscript, in the submitted form it is suitable for publication.